# Investigation of Biofouling and Its Effect on the Properties of Basalt Fiber Reinforced Plastic Rebars Exposed to Extremely Cold Climate Conditions

**DOI:** 10.3390/polym14030369

**Published:** 2022-01-18

**Authors:** Anatoly K. Kychkin, Larisa Anatoljevna Erofeevskaya, Aisen Kychkin, Elena D. Vasileva, Nikolay F. Struchkov, Mikhail P. Lebedev

**Affiliations:** 1V.P. Larionov Institute of Physical and Technical Problems of the North, Siberian Branch of the Russian Academy of Sciences, 1 Oktyabrskaya Street, 677980 Yakutsk, Russia; vasilyeva_edm@mail.ru (E.D.V.); Struchkov_n@rambler.ru (N.F.S.); 2Institute of Oil and Gas Problems, Siberian Branch of the Russian Academy of Sciences, 677000 Yakutsk, Russia; lora-07.65@mail.ru; 3Federal Research Center “The Yakut Scientific Centre of the Siberian Branch of the Russian Academy of Sciences”, 2 Petrovskogo Street, 677000 Yakutsk, Russia; icen.kychkin@mail.ru (A.K.); m.p.lebedev@mail.ru (M.P.L.)

**Keywords:** polymer composite materials, basalt-plastic reinforcement, microorganisms, cultures, cryophilicity, destructors, strength, porosity

## Abstract

For the first time, the possibility of penetration of mold fungi mycelium and spore-forming bacteria into the structure of basalt fiber reinforced plastic rebars has been shown in laboratory and field experiments. Biological contamination at the “fiber-binding” border reveals areas of swelling and penetration of mold fungi mycelium and bacterial spore cells into the binder component. After the exposure of samples at extremely low temperatures, strains of mold fungi of the genus *Aspergillus* were also isolated from the surface of the rebars. Additionally, spore-forming bacteria of the genus *Bacillus* immobilized for samples from two years ago. This indicates the high viability of immobilized strains in cold climates. Aboriginal microflora isolated by the enrichment culture technique from the samples was represented by: actinobacteria of the genera *Nocardia* and *Streptomyces*; yeast of the genus *Rhodotorula*; and mold fungi of the genus *Penicillium*. It was shown that the enrichment culture technique is a highly informative method of diagnosing the bio-infection of polymer composite materials during their operation in extremely low temperatures. The metabolic activity of the cells of cryophilic microorganisms isolated from experimental samples of basalt fiber reinforced plastic rebars was associated with the features of the enzymes and fatty acid composition of the lipid bilayer of cell membranes. In the case of temperature conditions when conventional (mesophilic) microorganisms stop developing vegetative cells, the process of transition of the lipid bilayer of cell membranes into a gel-like state was activated. This transition of the lipid bilayer to a gel-like state allowed the prevention of crystallization and death of the microbial cell when the ambient temperature dropped to negative values and as a result, after thawing, growth resumed and the metabolic activity of the microorganisms was restored. Studies have been carried out on the effect of biodepletion on the elastic strength characteristics, porosity and monolithicity of these materials, while at the same time, after a two year exposure, the strength preservation coefficient was k = 0.82 and the porosity increased by more than two times. The results show that the selected strains affect the properties of polymeric materials in cold climates in relation to the organic components in the structure of polymer composites.

## 1. Introduction

Damage from corrosion and aging increases due to biological damage to materials in a wide variety of operating conditions (atmosphere, hydrosphere, lithosphere, outer space). In many cases, biocorrosion is the dominant cause of the destruction. Currently, the study of the biological impact on polymer composite materials (PCM) is one of the key areas not only in microbiological science but in a number of other fundamental and applied disciplines related to the study of the properties of materials and their production [1,2,3,4].

Laboratory and natural tests search for new environmentally friendly biocidal additives and ways to protect metals, nonmetals and petroleum products from exposure to biopests [5,6,7,8].

There has been more evidence on microbial contamination and the impact of this process on the destruction of PCM. The incidence of microorganisms is most significant in geographical areas with a relatively high air temperature, high humidity, and abundance of organic dust (tropics and subtropics) [9,10,11,12].

The effect of microorganisms on polymeric samples causes their biodegradation to varying degrees. This is due to both the composition of polymeric materials and the different activity of different types of microorganisms with highly active extracellular hydrolases, phosphatases and other enzymes. First of all, these are fungi from the genera *Aspergillus*, *Penicillium*, *Trichoderma*, *Cladosporium*, *Fusarium* and bacteria of the genera *Pseudomonas*, *Streptomyces*, *Bacillus*, and *Arthrobacter* that destroy polymer compounds under oxidation conditions [13,14]. This accounts for only a small part of the studied destructors since cells that can grow in culture under laboratory conditions account for less than 1% of the total permafrost community of microbes. For the majority of viable cells, cultivation modes have not yet been found, and they are studied using culture-independent methods [15]. Nevertheless, a large number of studies are currently being conducted in the field of biodegradation. Given the huge metabolic potential of microorganisms, it is expected that the development of cost-effective and viable biodegradation processes is only a matter of time [16].

The choice of basalt fiber as a fibrous reinforcing component among various polymer composite materials (PCM) is based on such positive properties as high strength, heat resistance, resistance to chemically active compounds [17,18,19,20,21,22], fire resistance, inertness to the action of mold fungi and microorganisms [23], abrasion and shock resistance [24].

However, the question of the role of contamination of materials by cryophilic microorganisms and changes in the properties of PCM under the influence of biological effects during their operation in the climatic conditions of the Far North remains poorly understood [25]. The topic of biofouling of promising basalt fiber reinforced plastic (BFRP) rebar is virtually unexplored, including in the northern regions. There is no information on the participation of destructing microorganisms in the processes of biodeterioration of the BFRP rebars.

The ability of a cryophilic group of microorganisms to adsorb materials from the external environment depends largely on the adhesive properties of microbial cells. Adhesive properties are characteristic of many cryophilic bacterial associations and microscopic molds capable of growing at temperatures from 4–5 to 15–20 °C. Under favorable conditions (temperature, humidity, pH, etc.), the processes of adhesion are replaced by the process of penetration of microorganisms into microcracks. They accumulate their biomass, resulting in internal stresses that lead to the expansion and branching of microcracks. As a result, deterioration of the strength properties of PCM occurs [26,27]; however, the adhesion and contamination ability of bacteria and fungi at temperatures below minus 40 °C is practically unexplored.

It is worthwhile investigating the issue of bio-infection of BFRP rebar by cryophilic microorganisms and studying the mechanism of biodamage of basalt fiber reinforced plastic rebars in an extremely cold climate. This will allow us to create a database of microbiological agents of biodeterioration of PCM during their long-term use in environmental conditions. The obtained results can further serve as the basis for the development of compositions for the prevention of biological damage to BFRP rebar and other polymer composite materials (PCM) during long-term storage, operation and for inactivation of biodestructive microflora, which can be stored for hundreds of years buried in the earth and can be discovered and carried by air during construction and earthworks, settling on surfaces, causing the biocontamination and biodeterioration of construction materials, in particular BFRP rebar.

The purpose of this investigation was to identify and study the cultural and biochemical characteristics of cryophilic microorganisms capable of bioinfection of PCM, in particular BFRP rebar, at negative temperatures in an open ecosystem and the impact of this process on changes in the properties of the BFRP rebar.

## 2. Materials and Methods

### 2.1. Raw Materials

The reinforcing member was RBN 13-2400-4S basalt roving with a 13 microns diameter fiber (“TBM” LLC, Yakutsk, Russia). The nominal linear density of roving was 2400 tex. The breaking load of the roving was not less than 320 mN/tex. The binder was an epoxy resin ED-22 (Kukdo chemical Co., Ltd., Seoul, Korea), which was “hot” cured with a suitable amount of hardener in the presence of an accelerator. The hardener, isomethyltetrahydrophthalic anhydride (iso-MTHPA) (JSC Sterlitamak Petrochemical Plant, Sterlitamak, Russia), was chosen because the systems cured with it have high mechanical properties, excellent waterproofing properties, good electrical properties, and resistance to climatic impact. A 2,4,6-tris(dimethylaminomethyl)phenol (UP 606/2) accelerator was chosen due to its high catalytic activity (JSC Sterlitamak Petrochemical Plant, Sterlitamak, Russia).

### 2.2. Composite Specimen Preparation and Characterization

Specimens of BFRP rebar with diameters of 6, 8, 12, and 20 mm, and with a height of 1000 mm were manufactured. The main stages of manufacturing the BFRP rebar included: continuous basalt roving (bundled thin basalt fibers with a diameter of 9...13 microns) impregnated with thermosetting epoxy resin; some of the impregnated rovings were pulled through the central spinneret forming a forced (8 kgF) inner rebar with a diameter of 4, 8 mm, respectively, then this rebar was cured; some of the impregnated rovings were pulled through the peripheral spinneret forming an outer layer and fixed to the cured inner rebar and after that were wrapped with a bundle of polyamide yarns to increase the adhesion of rebar with mating materials (concrete, lime, cement mortars, etc.).

The BFRP rebar was prepared in accordance with the *Technical Specifications 2296-001-86166796-2013 “Non-metallic composite armature made from basalt plastic”*.

The profile and key of the rebar sizes are shown in Figure 1. The rebar with the diameter d, with a step t, and wavy protrusions of height h rise, forming the outer diameter of the bar d1, is according to Table 1.

### 2.3. Climatic Aging of BFRP Rebar

The biofouled BFRP rebars were exposed at open stands in the extremely cold climate of the city of Yakutsk for 24 months (Figure 2). Yakutsk is located in a zone of sharply continental climate with very cold winters and relatively hot and short summers. The minimum recorded temperature is minus 64.4 °C. The annual temperature amplitude between the highest and lowest values exceeds 100 °C. The average wind speed is 1.8 m/s. The average annual relative humidity in Yakutsk is 68%. During the year, 237 mm of precipitation falls in Yakutsk. The annual total solar radiation dose in Yakutsk is 3680 MJ/m^2^.

Table 2 shows the climatic characteristics of the Yakutsk city.

### 2.4. Methods of Studying Biofouling

The study of the impact of biofouling on the properties of BFRP rebar at low temperatures was carried out in three stages.

At the first stage, rinse sampling, the enrichment culture technique in the mineral medium and native application of the test samples on nutrient agar were used to isolate microorganisms.

For rinse sampling, buffered peptone water was preliminarily prepared with the following composition (gram/liter): proteose peptone-10.00; sodium chloride-5.00; sodium hydrogen phosphate-3.50; and potassium dihydrogen phosphate-1.50. Buffered peptone water was used as a means to increase the efficiency of the diagnostics of BFRP rebar biofouling by cryophilic microorganisms.

The ingredients of the above-described composition were stirred in 1 L of distilled water, after which the pH was adjusted to 7.2 ± 0.2 conventional units and poured into sterile glass bottles of 50 mL.

Vials of the buffered peptone water were sterilized at 1.1 atmosphere (121 °C) for 15 min.

After sterilization, the buffered peptone water vials were cooled at room temperature.

Then, the required number of pre-sterilized biological test tubes with cotton swabs were prepared for taking rinse samples from the BFRP rebar samples No. 1–4, exhibited at the climatic test site.

In the prepared test tubes with cotton swabs, 2 mL of the cooled sterile buffered peptone water were aseptically poured. Then, the BFRP rebar samples were swabbed, and the samples were immediately brought to the laboratory for microscopic examination for the preliminary express diagnostics for bio-infection.

To obtain the enrichment culture of the microorganisms, rinsing liquid residue from BFRP 1–4 samples in the above method were incubated within 24 h at room temperature. After that, they were moved to a meat-peptone agar (MPA) and an industrially prepared nutrient agar of Saburo and Chapeka (The State Research Center for Applied Microbiology, Obolensk, Russia).

To isolate the cryophilic microorganisms, enrichment of the cultures on the prepared nutrient media was carried out in stationary conditions in the refrigerator at 4 ± 1 °C.

To diagnose bio-infection by the native method, about 6 mm thick layers of nutrient agar were poured into sterile Petri cups or containers of the required size. After hardening, they were dried in a thermostat TS-80 (Kasim Instrument Plant, Kasimov, Russia) at 37 °C for 40 min. The nutrient agar was dried using the method described in “On Unification of Microbiological (Bakteriological) Methods Used in Clinical Laboratories of Health Care Facilities”. After that, the prepared samples were laid out on the surface of the hardened nutrient medium in the form of rebars 20 cm long, pre-prepared from the sample No. 4. The containers were covered with lids, placed in the refrigerator and incubated at 4 ± 1 °C to record the growth of the cryophilic microorganisms and at 25, 30 and 37 ± 1 °C to record mesophilic microorganisms for 5, 3, and 1 day, respectively.

To isolate the pure cultures and study their cultural and biochemical properties, the resulting colonies were re-inoculated in Petri dishes on the surface of the MPA with subsequent re-inoculation on agar slanted in biological test tubes and biochemical tests.

To determine the relationship of microorganisms and to build phylogenetic trees, the sequences were processed using software from the RDP-II website (Ribosomal Database Project II, Michigan State University, East Lansing, MI, USA) and conventional methods of genetic identification [28,29,30,31].

In the second stage, in order to study the process of biodamage of the BFRP rebars by the microorganisms, the 20 cm long test samples of warehouse storage No. 4 were placed in the soil at a depth of 20 cm. The experimental exposure lasted 1 year.

A year later, the samples were removed from the soil and microbiological studies of rinsed material taken from the BFRP rebars were carried out. The separation of microorganisms was carried out in the ways described above, followed by identification.

A working collection was formed from the dominant cultures that is promising for biotechnological applications and studying selected microorganisms as PCM destructors.

At the third stage, biological products were obtained from the working collection of microbial cultures, which were used to cover the BFRP rebars warehouse storage and 1000 mm long contaminated samples were placed for exposure at the climatic test site.

### 2.5. Monolithic Test

The monolithic test method was carried out in accordance with the Russian standard GOST 32486-2015. The test method was designed to assess the integrity of composite reinforcement and establishes the procedure for determining the penetration of liquid with dye in its longitudinal direction. Longitudinal porosity was determined by controlling the penetration of the dyed liquid into the composite reinforcement. Penetration occurs as spots or points displayed on the opposite dry end of the sample. A 0.25% fuchsine solution dissolved in ethyl alcohol according to the standard GOST 18300 was used as a penetrating liquid. Samples No. 4 of 20 mm diameter for warehouse storage after a 28 month exposure at the test site and biofouled ones after a 24 month exposure, were compared. Five samples 5–6 mm long were cut out from the BFRP rebars. The samples were placed in the Petri bowl with the end down on a thin mesh substrate laid on the bottom of the Petri dish in order to fix the sample, evenly moisturize the lower surface of the samples and reduce air involvement. The dye was poured so that the lower end was covered with dye by 1–2 mm. The time was measured until the first spot appeared on the opposite end of the test sample.

### 2.6. Porosity Studies

To determine the open porosity, the method of hydrostatic weighing was used. Due to its high penetrating power, kerosene was used as an impregnating liquid. To carry out the test, samples with lengths of ~30 mm and 5 pieces of each diameter, were cut from the BFRP No. 1 from the warehouse storage and exposed rebars. Before testing, the samples were pre-dried in a ShSV-65 (3.5) vacuum drying cabinet (CJSC MIUS, Tula, Russia) at a temperature of 50 °C for 5 h. To weigh the samples in air and in kerosene, an analytical balance Vibra HTR-220CE (Vibra, Tokyo, Japan) with a measurement error of 0.1 mg and range of 0.01–220 g, accuracy class I.

According to the method of hydrostatic weighing, the formula for determining the open porosity of the coating is as follows:(1)$$P_{0}=\frac{m_{2}-m}{m_{2}-m_{1}}$$
where *m* is the mass of a dry sample in air; *m*_1_ is the mass of the impregnated sample in the liquid; and *m*_2_ is the mass of the impregnated sample in air.

To carry out the measurements to determine porosity, the following steps must be taken:Sample preparation.Measurement of dry mass (*m*) of samples.Impregnation of samples in kerosene for 1 day.Weighing of impregnated samples (*m*_2_) in air.Weighing of impregnated samples (*m*_1_) in kerosene.

### 2.7. Macroanalysis of Material Structure

Micrographs of the samples were obtained using:(1)Polarizing microscope Axiolab Pol (Carl Zeiss, Jena, Germany) at ×1000 magnification.(2)Scanning electron microscope G1200 12MP 1-1200X (KKMOON, Huanan, China) at ×10 magnification.

### 2.8. Tensile Tests

Tensile tests were carried out on the Z600 Zwick/Roell (Zwick, Ulm, Germany) universal test machine according to the standard GOST 32492-2015 “Composite polymer reinforcement for concrete structures”, “Methods for determining physical and mechanical characteristics”.

Tensile tests were carried out at room temperature +20 °C, at a relative humidity of 40%. The calculated length of the sample during the tests was 200 mm. The preload speed was 10 mm/min. Test speed was 5 mm/min.

BFRP No. 1 of a 6 mm diameter from warehouse storage after 28 month exposure at the test site and biofouled ones after a 24 month exposure were used for comparative analysis.

## 3. Results and Discussion

### 3.1. Isolation of Microbial Cultures from Rinse Probes

Three groups of cryophilic microorganisms were isolated from samples of BFRP No. 1–4 exhibited in winter in an open climate test site at extremely low temperatures: bacteria, actinobacteria and microscopic mold fungi (Table 1). On the day of material selection (December 22) for the microbiological studies, the air temperature was −42 °C.

The received rinse material was planted on the surface of the nutrient agar, then it was incubated in a refrigerator at a temperature of plus 4 ± 1 °C. Already after seven days of incubation on the plates with nutrient media, a full growth of microorganisms was noted, which were identified by cultural-biochemical, morphological and microscopic methods (Table 3).

Bacteria of the genus *Bacillus*, actinobacteria of the genus *Nocardia* and microscopic mold fungi of the genera *Aspergillus* and *Fusarium* were isolated from fragments of the BFRP rebars by the above method of culture accumulation.

It was established that the share of viable cells in the rinse fluid was 33%. The isolated microorganisms are distributed in the following ratio: bacteria–23% and mold fungi–33%, actinobacteria capable of forming substrate and air mycelium–44%.

At the same time, the method of native application of the BFRP No. 4 test samples onto nutrient agar was studied (Figure 3).

Typical fouling of the BFRP No. 4 by colonies of *Bacillus subtilis* and *Streptomyces albus* in agar-based medium after 72 h. Scale 1:1.

The qualitative composition of microflora isolated by the native method was represented by bacteria of genera *Bacillus*, and *Pseudomonas*; actinobacteria of genera *Nocardia*, and *Streptomyces* and microscopic mold fungi of genera *Aspergillus* and *Fusarium* (Table 3).

The landscape of microorganisms contaminating the surface of the BFRP rebar fragments was dominated by cryophilic species of spore-forming bacteria: *Bacillus subtilis*, and *Bacillus cereus*; non-enzymatic bacteria *Pseudomonas aurugenosa*; actinobacteria *Streptomyces albus* and *Nocardia* sp.; and mold fungi: *Aspergillus niger*, *Aspergillus fumigatus* and *Fusarium* sp.

The biochemical properties of the isolated microbial cultures are presented in Table 4.

The microscopic method showed that the cell size of bacillary strains from the surface of the BFRP rebar test samples was 0.7–3.0 µm for *Bacillus*; 0.6 × 1.5 µm for *Pseudomonas*; 1–2 µm (hyphal diameter) and 0.5–2.0 µm (spore diameter) for *Streptomycetes*; 0.5–2.0 µm (hyphae diameter) for *Nocardia*; 2.5–5.0 µm (diameter of hyphae of aerial mycelium), 8–12 µm (chlamydospores), 30 × 5 µm (macroconidia) for *Fusariums*; 4–6 µm (mycelium width), 7–10 µm (hyphae), 2–3.5 µm (spores) for *Aspergillus*.

### 3.2. Effect of Mold Fungus on Change of the BFRP Rebar Structure

Based on the selected mold fungus cultures, *Aspergillus niger* and *Aspergillus fumigates,* in a ratio of 1:1 with a concentration of at least 1 × 10^9^ cells per 1 cm^3^, liquid biological products were made, which were used to process the rebars of BFRP samples in order to study their biodamaging properties.

Samples 1000 mm long of basalt plastic rebars prepared for the experiment were infected with a liquid biological product by immersion in a bath following the standard, GOST R MEC 60068-2-10-2009 “Tests on external factors. Part 2–10. Tests. J test and manual: Fungal resistance.” The method is easy to use and promotes better adhesion of spores to the surface of the contaminated material. After infection with spores, the rebars of the BFRP rebar were dried by the contact method and exhibited on the territory of the open climate test site in Yakutsk for 12 months.

A year after exposure, the genus *Aspergillus* was discovered in a sample of BFRP rebars by microscopy, which indicates their survival in cold climates.

The growth of mycelium was recorded both in the direction along the length between the basalt fibers at the boundaries of the “fiber-binding (hardener)-mycelium of fungi-fiber”, and across the fibers. Additionally, microscopy showed rebar-shaped bacterial cells on the BFRP rebar samples. Some of them had spores at one end, which indicates the preservation of microorganisms within the BFRP rebars after exposure at extremely low temperatures, to minus 42 °C on the day of sampling (Figure 4 and Figure 5).

The length of the fungal mycelium, which grew through test samples in one year, was recorded at the level of 1.4 to 1.8 μm in microscopic examination, which in terms of the 1 g of sample BFRP rebar under study was about 0.035–0.045 cm or 0.00035–0.00045 m. The recalculation was made using the Johnson and Mollis method [32].

At the second stage, in order to study the process of biodamage of the BFRP rebars by microorganisms, test samples were placed in the soil at a depth of 20 cm. The experimental exposure time was also one year (Figure 6).

One year after the removal of samples from the soil, during visual control, the beginning of the destruction process under the influence of mold fungi on the samples had a form of dimming caused by the degradation of the organic components of the polymer matrix.

The microscopic method in the dimming zone on damaged areas of the BFRP rebars recorded simmering of the binder inside the samples (Figure 7).

Polarizing microscopy with a ×1000 magnification showed that areas of binder swelling and rebar-shaped spore cells of bacteria (Figure 7) were visible at the border of the “fiber-binding”, which probably penetrated the basalt plastic rebars from environmental objects, in particular from the soil. Spore-forming bacteria of the genus *Bacillus* were isolated from the surface of the samples and the soil taken from the site of the experiments (*B*. *subtilis*, *B*. *simplex*, *B*. *vallismortis*, *B*. *cereus*), which confirmed the likelihood of contamination of the rebars by soil bacteria.

From the dominant bacterial and fungal cultures identified during the research, a working collection has been formed that is promising for biotechnological use and studying the selected microorganisms as potential PCM destructors. Maintaining the viability of microorganisms at extremely low temperatures (up to minus 42 °C) and the resumption of their metabolic activity when entering favorable temperature and other conditions, is one of the common signs of cryophilicity for the strains under study, which allows the use of microorganisms selected for bioinfection and for studying the processes of destruction in samples outside the vegetation period (i.e., in winter or right before winter).

The temperature minimum relevant to the development of vegetative cells and spores of the microorganisms-destructors included in the working collection is presented in the following order: for bacteria of the genus *Bacillus* +3–5 °C (*B*. *vallismortis* +3 °C, *B*. *cereus* +3 °C, *B*. *simplex* +4 °C, *B*. *subtilis* +5 °C); and for mold fungi of the genus *Aspergillus* +3–4 °C (*A*. *niger* +3 °C, *A.*
*fumigates* +4 °C).

The metabolic activity of selective strain cells at minimum temperatures (+4 ± 1 °C) is associated with the features of enzymes and fatty acid composition of the lipid bilayer of cell membranes, which are in a liquid–crystalline state in conditions optimal for microorganisms. In the case of temperature conditions, when the development of vegetative cells temporarily stops—in particular from +3–5 °C for strains of the working collection—the process of transition of the lipid bilayer of cell membranes into a gel-like state is activated, which prevents crystallization and death of the microbial cell with a gradual decrease in ambient temperature to negative values, so that after thawing growth resumes and the metabolic activity of the microorganism is restored; however, a sharp transition from an optimal to extremely low ambient temperature can contribute to the destruction and damage of microbial cells. This can happen when refreezing and defrosting selected strains during experiments, or after samples or biologics after storage at room temperature or a temperature of +3–5 °C are sharply taken out for exposure in winter conditions at extremely low temperatures. The death of the immobilized cells of microorganisms on samples can occur due to impaired barrier functions of cell membranes.

### 3.3. Fungicidal Activity of Bacterial Strains from BFRP Rebars Samples

To study fungicidal activity, bacterial strains of the genus *Bacillus* (*B*. *simplex* and *Bacillus* sp.) were selected, in which a targeted study revealed fungicidal activity in relation to mold fungi of the genus *Aspergillus* (*A*. *niger* and *A*. *fumigates*).

Brief characteristics of the strains.

*Bacillus* simplex are Gram-positive rebar cells with centrally located ellipsoid spores. On the IPA forms shiny, wet, large colonies of creamy white color, stretching for a bacteriological loop.

Initial screening on the GenBank and RDP-II database showed that the strain under study belongs to the following systematic groups: *Bacteria*, *Firmicutes*, *Bacilli*, *Bacillales*, *Bacillaceae*, *Bacillus*.

The results of sequence processing using software located on the RDP-II (Ribosomal Database Project II) website, designed to determine the relationship of microorganisms and build phylogenetic trees, are presented in Table 5.

Analysis of phylogenetic kinship, built using typical strains of closely related bacteria, taking into account the morpho-physiological properties of the strain, shows that the closest to the strain under study is the species *Bacillus simplex* (100%).

*Bacillus* sp.-Gram-positive rebar-shaped cells with centrally located oval spores. On the IPA forms matte, large, opaque, grayish colonies with an uneven, rough surface and uneven edges with a diameter of up to 5 mm or more.

Initial screening on the GenBank and RDP-II database showed that the strain under study belongs to the following systematic groups: *Bacteria*, *Firmicutes*, *Bacilli*, *Bacillales*, *Bacillaceae*, and *Bacillus*.

The results of sequence processing using software on the RDP-II website (Ribosomal Database Project II), designed to determine the relationship of microorganisms and build phylogenetic trees, are presented in Table 6.

Analysis of phylogenetic kinship, built using typical strains of closely related bacteria, taking into account the morpho-physiological properties of the strain, shows that three types of cryophilic strains of the genus *Bacillus* are closest to the strain under study: *B*. *altitudinis* (99.89%); *B*. *stratosphericus* (99.89%); and *B*. *safensis* (99.89%). According to this criterion, the bacterial strain under study is classified as *Bacillus* sp. (99.89%).

The presence of fungicidal activity in the studied bacterial strain was judged by the presence of an inhibition zone of the test strain.

The results of the tests of the selected strains for antagonistic activity are presented in Table 7.

It has been established that when testing the relationship between antagonist strains and mold-sensitive test cultures, antagonistic activity manifests itself higher when using a paired composition of cultures (*Bacillus* sp. + *Bacillus simplex* and *Aspergillus niger* + *Aspergillus fumigatus*) than when using the same microorganisms as part of monocultures.

At the same time, fungicidal activity to selected mold fungi was studied. The experiment showed that both bacterial cultures of the genus *Bacillus* (*B*. *simplex* and *Bacillus* sp.) have fungicidal activity to strains of mold fungi of the genus *Aspergillus* (*A*. *niger* and *A*. *fumigatus*) (Table 8).

A common established feature for the strains under study is that they, having cryophilic properties, can develop at low temperatures in conditions of limited oxygen access to the environment. This indicates that selected strains can show their destructive properties in cold climates relative to organic components in the structure of polymer composites when their metabolism is slowed down and there is no air exchange. It is shown that the method of enrichment culture is a highly informative method of diagnosing the bio-infection of polymer composite materials during their operation in extremely low temperatures.

### 3.4. Effect of Biofouling on Physico-Mechanical Properties of BFRP Rebar

In order to study the participation of selected microorganisms in the biodestruction of organic components of PCM in low ambient temperatures, liquid forms of biological products were manufactured, which were used to treat the BFRP rebar of warehouse storage.

Two types of biological product were used in the experiment: the first was based on *Bacillus* bacteria and the second was based on mold spores of the genus *Aspergillus*.

The concentration of cells in the biologics at equal ratios (1:1) was 1–9 × 10^9^ CFU/cm^3^ for the fungi and bacteria. Exposure of the bio-infected BFRP rebars was carried out in accordance with the standard, GOST 9.708-83, in an extremely cold climate in the city of Yakutsk, at the climate station of the Institute of Physical and Technical Problems of the North, Siberian Branch of the Russian Academy of Sciences, for two years.

After exposure time, rinses from the BFRP rebar samples were selected for microbiological studies and tensile and porosity tests were carried out.

It has been found that bacteria of the genus *Bacillus* remained immobilized on all 43 samples; and strains of mold fungi of the genus *Aspergillus* were isolated from 10 samples of BFRP rebar, which measured 23%.

Aboriginal microflora isolated from the rinse fluid taken from the surface of test samples was presented by: actinobacteria of the genera *Nocardia* and *Streptomyces*; yeast of the genus *Rhodotorula*; and mold fungi of the genus *Penicillium*.

A set of phenotypic signs and antibiotic resistance of the experimental strains of bacteria and mold fungi served as a marker to confirm the identity of the BFRP rebar immobilized for the samples and isolated from the rinse fluid after two years of exposure.

Table 9 and Figure 8 show the results of the monolithic studies.

The result of the monolithic studies shows (Figure 8) that on the right side of the BFRP rebar, which was exposed to solar radiation, the penetration of the dye was more intense, filling the surface layers of the degraded surface layers. In the bio-infected samples, the time until the first point of dye appeared on the end of the sample was four times less, which shows a higher dimension of the formed pores due to the penetration of mold mycelia.

To study the change in strength properties of the rebars during their biodegradation under short-term tension, rebar specimens with a diameter of 6 mm were tested at an ambient temperature of 20 °C and a relative humidity of 40–50%.

To determine the climatic and biogenic impact, the change in the generalized indicator—the relative coefficient of conservation—was calculated:(2)$$K_{R}=\frac{R_{t}}{R_{0}}$$where *R_t_* is the ultimate tensile strength (*σ_t_*) and porosity (*P_o_*), measured after different periods of exposure; *R*_0_—initial values of the corresponding indicators. The results of comparative tests of BFRP No. 1 for axial tension and open porosity are shown in Table 10.

When comparing the values of the ultimate tensile strength for the original samples and biologically contaminated samples exposed for 24 months, one can see a decrease in the ultimate strength by 18% and an increase in porosity by more than two times in a shorter period.

The received results show that the selected strains affect the properties of polymeric materials in cold climates in relation to organic components in the structure of polymer composites.

## 4. Conclusions

As a result of microbiological studies from the surface of samples of basalt plastic composite materials exhibited at the climatic test site in an open ecosystem at extremely low ambient temperatures (up to minus 42 °C), various types of cryophilic microorganisms have been identified; the share of viable cells was 33%, of which were bacteria–23%; mold fungi–33%; and actinobacteria–44%. The metabolic activity of cryophilic microorganism cells isolated from the experimental samples of BFRP rebars is associated with the features of enzymes and fatty acid composition of the lipid bilayer of cell membranes. In conditions optimal for life, microorganisms are in a liquid–crystalline state, and in the case of temperature conditions, when ordinary (mesophilic) microorganisms stop developing vegetative cells, the process of transition of the lipid bilayer of cell membranes into a gel-like state is activated. This allows the prevention of crystallization and death of the microbial cell when the ambient temperature drops to negative values, so that after thawing growth resumes and the metabolic activity of the microorganism is restored.In samples of polymer composites treated with microbial spores, after one and two years of exposure at the climate test site, cells of spore-forming bacteria of the genus *Bacillus* and mycelium of mold fungi of the genus *Aspergillus* were found, which indicates the survival of immobilized microorganisms in a cold climate. In laboratory and field experiments, for the first time, the possibility of penetration of fungal mycelium into the structure of BFRP rebars during their exposure in an open polygon was established. Biological contamination at the “fiber-binding” border reveals areas of blistering and penetration of bacterial spore cells into the binder component.Microbiological studies have found that after two years of exposure in extremely low temperatures, *Bacillus* bacteria remained immobilized on all 43 samples; and strains of mold fungi of the genus *Aspergillus* were isolated from 10 samples of BFRP rebars, which was 23% of all microflorae preserved in a viable state.It has been established that the bacterial strains *Bacillus* sp. and *Bacillus simplex* have fungicidal activity in relation to the strains of mold fungi *Aspergillus fumigatus* and *Aspergillus niger*. A common established feature for the strains under study is that they, having cryophilic and osmotolerance properties, can develop at low temperatures in conditions of limited oxygen access in the environment. This indicates that selected strains can show their destructive properties in cold climates relative to the organic components in the structure of polymer composites when their metabolism is slowed down and there is no air exchange. It was shown that the enrichment culture technique is a highly informative method of diagnosing the bio-infection of polymer composite materials during their operation in extremely low temperatures.The results of the studies have expanded the range of microorganisms capable of contaminating polymer composite materials at extremely low temperatures (−42 °C). The formation of a working collection of destructors of polymer composites has begun. The reliability of the taxonomic belonging of most microbial strains included in the working collection is confirmed by a modern molecular genetic method based on the nucleotide sequence of 16S rRNA of the gene.As a result of comparative tests, an impact on the strength properties of basalt fiber reinforcement plastic rebar by cryophilic mold fungi has been established when exposed in cold climates. There was a decrease in the strength limit by 18% and an increase in porosity by more than two times.The results show that the selected strains affect the properties of polymeric materials in cold climates in relation to organic components in the structure of polymer composites.

## Figures and Tables

**Figure 1 polymers-14-00369-f001:**
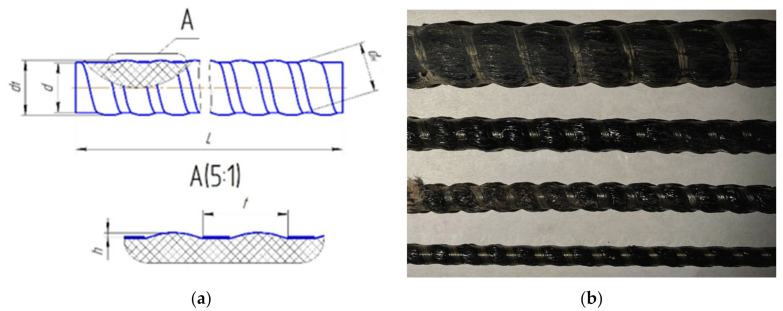
(**a**) The keys of periodic rebar profile, (**b**) samples of BFRP rebar.

**Figure 2 polymers-14-00369-f002:**
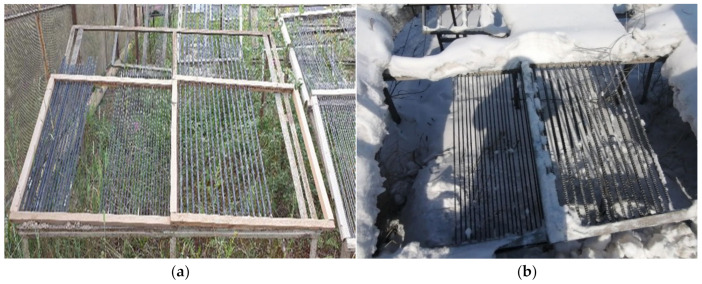
Exposure of BFRP rebar in Yakutsk: (**a**) in summer and (**b**) winter.

**Figure 3 polymers-14-00369-f003:**
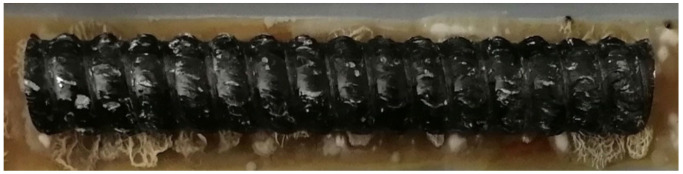
Picture of a typical fouling of BFRP No. 4 by colonies of *Bacillus subtilis* and *Streptomyces albus* in agar-based medium after 72 h. Scale 1:1.

**Figure 4 polymers-14-00369-f004:**
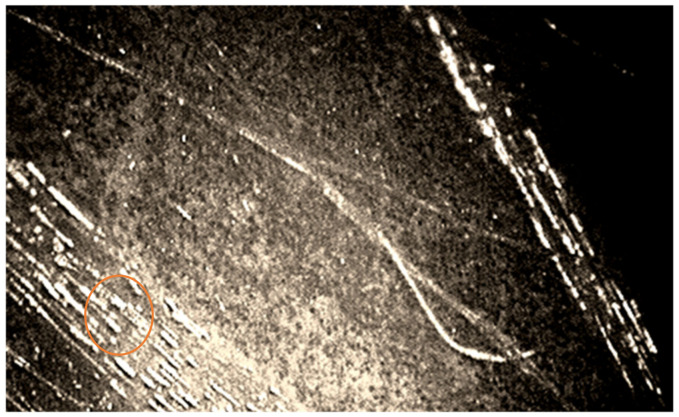
Image of the mycelium of fungi of the genus *Aspergillus* at the border “fiber-binder (hardener)-fungal mycelium-fiber” and cells of bacteria of the genus *Bacillus*, polarizing microscopy, ×1000 magnification.

**Figure 5 polymers-14-00369-f005:**
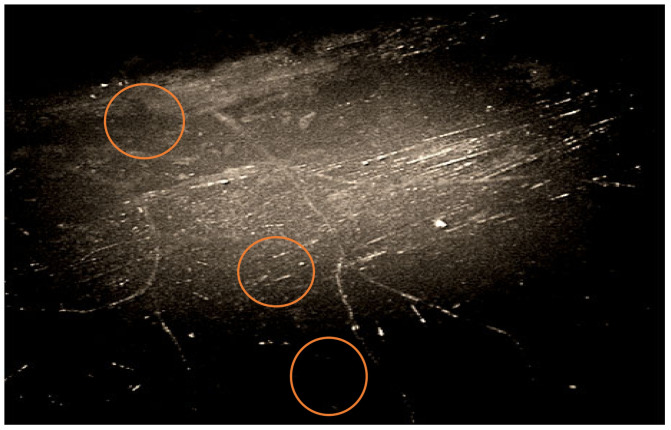
Image of the mycelium of fungi of the genus *Aspergillus*, germinating from the wall of the BFRP rebar and the cells of bacteria of the genus *Bacillus*, biologically infected sample BFRP rebar after climatic test, polarizing microscopy, ×1000 magnification.

**Figure 6 polymers-14-00369-f006:**
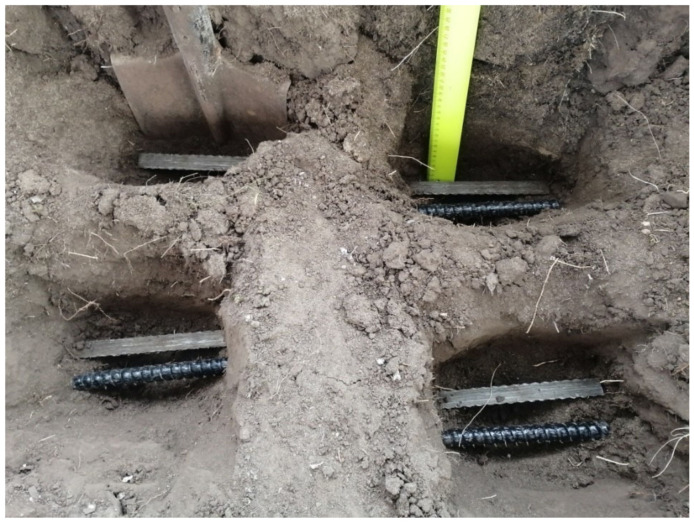
Picture of the field experiment configuration on the effect of destructive microorganisms on the process of biodamage of BFRP rebar test samples.

**Figure 7 polymers-14-00369-f007:**
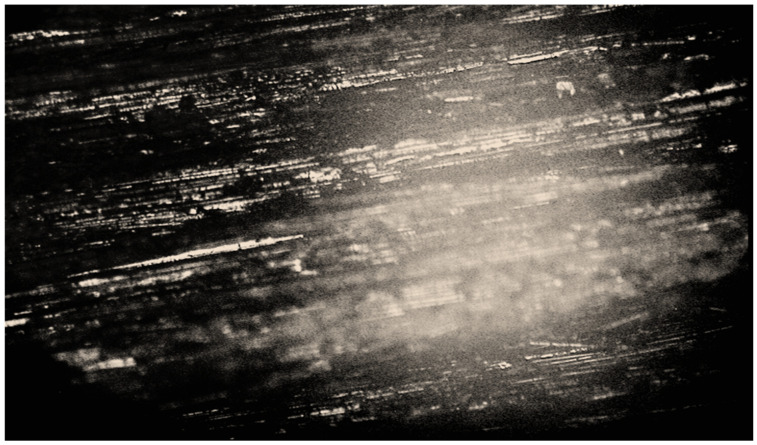
Image of a swelling of the binder in the damaged area of the BFRP rebar sample in contact with the soil, polarizing microscopy, ×1000 magnification.

**Figure 8 polymers-14-00369-f008:**
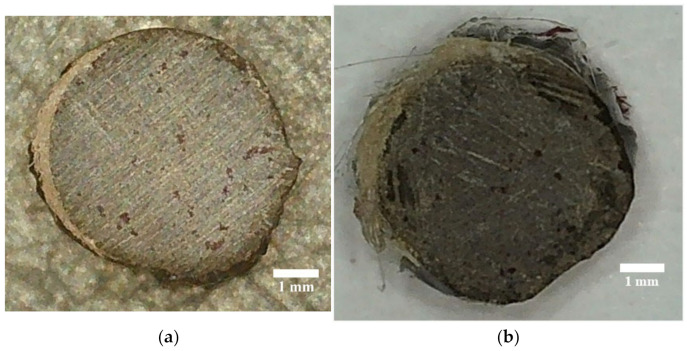
Images of BFRP rebar after testing for monolithicity with a diameter of 6 mm, with the original exposed for 51 months (**a**) and the bio-infected sample after exposure (**b**) for 24 months.

**Table 1 polymers-14-00369-t001:** Technological modes of forming the periodic profile of the rebar.

Sample	d (mm)	d1 (mm)	dh (mm)	h (mm)	t (mm)	Fibers Mass Share (%)
BFRP No. 1	6	7	6.0 ± 0.5	0.6	8.0	77
BFRP No. 2	8	9.0	8.0 ± 0.5	0.6	8.0	78
BFRP No. 3	12	13.0	12.0 ± 0.5	0.6	10.0	76
BFRP No. 4	20	21.0	20.0 ± 0,8	0.8	12.0	75

**Table 2 polymers-14-00369-t002:** The climatic characteristics of Yakutsk.

Month	Jan	Feb	Mar	Apr	May	Jun	Jul	Aug	Sept	Oct	Nov	Dec	Year
Average Solar Radiation per month, h	19	99	233	273	304	333	347	273	174	105	60	9	2228
Relative Humidity, %	75	75	70	61	54	57	61	67	71	77	77	75	68
Wind speed, m/s^−1^	0.8	0.9	1.5	2.2	2.6	2.5	2.3	2.2	2.2	1.9	1.3	1.0	1.8
The absolute maximum temperature, °C	−5.8	−2.2	8.3	21.1	31.1	35,1	38.4	35.4	27.0	18.6	3.9	−3.9	38.4
The average maximum temperatures, °C	−35.1	−28.6	−12.3	1.7	13.2	22.4	25.5	21.5	11.5	−3.6	−23.1	−34.3	−3.4
The average temperatures, °C	−38.6	−33.8	−20.1	−4.8	7.5	16.4	19.5	15.2	6.1	−7.8	−27	−37.6	−8.8
The average minimum temperatures, °C	−41.5	−38.2	−27.4	−11.8	1.0	9.3	12.7	8.9	1.2	−12,2	−31	−40.4	−14.1
The absolute minimum temperature, °C	−63	−64.4	−54.9	−41	−18.1	−5.4	−1.5	−7.8	−14.2	−40.9	−54.5	−59.8	−64.4
Rainfall, mm	9	8	7	8	20	35	38	37	31	18	16	10	237

**Table 3 polymers-14-00369-t003:** Viable microorganisms isolated from test samples.

Sample Name	Isolated Microorganisms	
	Rinse Method	Native Method
BFRP No. 1	*Aspergillus*	*Bacillus*
	*Nocardia*	*Aspergillus*
BFRP No. 2	*Fusarium*	*Bacillus*
	*Nocardia*	*Fusarium*
	*Bacillus*	*Streptomyces*
		*Nocardia*
BFRP No. 3	*Aspergillus*	*Bacillus*
	*Nocardia*	*Aspergillus*
BFRP No. 4	*Nocardia*	*Bacillus*
	*Bacillus*	*Aspergillus*
		*Pseudomonas*

**Table 4 polymers-14-00369-t004:** Studied properties of isolated microorganisms (+ positive test, − negative test, (+) weakly positive test).

Property (Test)	Dedicated Cultures					
	*Fuzarium* sp.	*B. cereus*	*Ps. aurugenosa*	*A. niger*	*Nocardia* sp.	*S. albus*
*Gram stain*	+	+	−	+	+	+
*Mobility*	−	+	+	−	−	−
*Glucose*	+	+	+	+	+	+
*Lactose*	+	−	−	+	+	−
*Sorbitol*	+	−	−	+	+	+
*Inositol*	−	+	−	−	+	−
*Maltose*	−	−	−	+	+	−
*Mannit*	+	+	+	+	+	−
*Sucrose*	+	+	−	+	+	+
*Catalase*	−	+	−	−	+	−
*Oxidase*	−	+	+	−	−	−
*Lecithinase*	−	+	−	+	+	−
*Sodium malonate*	+	+	−	+	+	+

**Table 5 polymers-14-00369-t005:** Phylogenetic identification of closely related bacteria to the studied strain (911 nt) *Bacillus simplex*.

Name	Strain	Authors	Accession	Pairwise Similarity (%)	Diff/Total nt
*Bacillus simplex*	NBRC 15720(T)	Priest et al. 1989	AB363738	100.00	0/911
*Brevibacterium frigoritolerans*	DSM 8801(T)	Delaporte and Sasson 1967	AM747813	99.89	1/911
*Bacillus muralis*	LMG 20238(T)	Heyrman et al. 2005	AJ628748	99.67	3/911
*Bacillus butanolivorans*	K9(T)	Kuisiene et al. 2008	EF206294	99.34	6/911
*Bacillus psychrosaccharolyticus*	ATCC 23296(T)	(ex Larkin and Stokes 1967) Priest et al. 1989	X60635	97.50	22/879

**Table 6 polymers-14-00369-t006:** Phylogenetic identification of closely related bacteria to the studied strain (922 nt) *Bacillus* sp.

Name	Strain	Authors	Accession	Pairwise Similarity (%)	Diff/Total nt
*Bacillus aerophilus*	28K(T)	Shivaji et al. 2006	AJ831844	99.89	1/922
*Bacillus altitudinis*	41KF2b(T)	Shivaji et al. 2006	AJ831842	99.89	1/922
*Bacillus stratosphericus*	41KF2a(T)	Shivaji et al. 2006	AJ831841	99.89	1/922
*Bacillus safensis*	FO-036b(T)	Satomi et al. 2006	AF234854	99.46	5/919
*Bacillus pumilus*	ATCC 7061(T)	Meyer and Gottheil 1901	ABRX01000007	99.35	6/922
*Bacillus idriensis*	SMC 4352-2(T)	Ko et al. 2006	AY904033	97.18	26/922

**Table 7 polymers-14-00369-t007:** Antagonistic activity of cultures (antagonist/test culture; + test positive; − test negative.).

Research Method	*Bacillus* sp./*A. niger*	*Bacillus* sp./*A. fumigatus*	*Bacillus* sp./*B. simplex*	*B. simplex*/*A. niger*	*B. simplex*/*A. fumigatus*	*A. niger*/*A. fumigatus*
Streak plate method	+	+	−	+	+	−

**Table 8 polymers-14-00369-t008:** Fungicidal activity of the selected strains (+ test is positive; − test is negative; (+) test is weakly positive).

Properties	Experimental Immobilization Cultures		Cultures Isolated from Fragments of a Sample	
	*Bacillus* sp.	*B. simplex*	*A. fumigatus*	*A. niger*
*A. niger*	+	+	−	−
*A. fumigatus*	+	+	−	−

**Table 9 polymers-14-00369-t009:** Results of testing the exposed samples for solidity before and after bio-infection and exposure in Yakutsk.

Mark	The Number of Dye Development at the End of The Sample, Number	Time until the First Point of the Dye Appears at the End of the Sample, sec
BFRP No. 1	80	20
Bio-infected BFRP No. 1	29	5

**Table 10 polymers-14-00369-t010:** Results of comparative tests of BFRP No. 1 for axial tension and open porosity.

Specimens	Parameter	Measure Value, MPa	K_R_-Coefficient of Property Retention
Initial samples	*σ_t_*	1120	0
	*P_o_, %*	0.26	0
After exposure in Yakutsk for 28 months	*σ_t_*	1206	1.08
	*P_o_, %*	0.36	1.38
After exposure bio-infected samples in Yakutsk for 24 months	*σ_t_*	920	0.82
	*P_o_, %*	0.57	2.19

## Data Availability

The data presented in this study are available on request from the corresponding author.

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
