# Peer review of "Investigation of Biofouling and Its Effect on the Properties of Basalt Fiber Reinforced Plastic Rebars Exposed to Extremely Cold Climate Conditions"

_polymers, 2022, doi:10.3390/polym14030369_

Round 1

Reviewer 1 Report

Dear Authors,

The manuscript 'Studying the effect of biofouling on the properties of basalt plastic rods at low temperatures' raises an interesting topic. The examined material is part of sustainable development. Therefore, testing its properties in various extreme conditions is crucial from the point of view of its further possible use. In this regard, the work is worth attention.

However, the paper is not sufficient for publication at this moment as it stands. The current content of this article does not reach the level of this Journal. This paper has potential, but significant points have to be clarified or fixed. The work fit for a rejection or major revision - most of the work has to be rewritten.

First of all, the topic does not reflect the content—a significant part of the manuscript descriptions and analysis of the fungi and moulds themselves. The title indicates an in-depth analysis of the properties of the test material subjected to changing environmental conditions, not the contamination itself.

Introduction - there is no information about the material over which the research will be carried out and no comparisons with research already conducted. More attention should be paid to the material than the experiment conditions.

Materials and Methods – This section has to be rewritten. The text does not provide any information on PCM, and the research aims to investigate the properties of basalt-plastic reinforcement BPR. Furthermore, it does not describe the experiment design: no detailed information on what and how many samples were tested. (The term "varius diameters" is unacceptable).

It is obligatory to write clearly for the reader what material was tested and under what conditions. Perhaps the diagram allows arranging the research layout legibly. When the tests start, when they end.

The specimen names representing the test groups should be given in this section.

Table 2, consists of the names of the samples 1 - 4; what they are? What do they represent? Whereas, Figure 4 caption has a sample name 3K2-2-3PO - which does not contribute to the article because we do not know what it means. The same with the Figure 5 caption.

The authors already published the article with a correctly written Material and Methods section, why this manuscript does not include similar.

Figure 2 shows what kind of sample? Maybe it is better to use  - "Image of representative basalt plastic rebar fouling with colonies of B ………."

Explain Figure 2 - whether some samples cut longitudinally and the other not, figure 2 shows cut sample. Why were they cut,  for what purpose?

Line 120 - Are you sure "thermostat" was used to dry the samples?

 Line 120 - cannot write "the prototype", and this is the end of the description. In addition, the term sample should be used. With an indication of the names of the samples, uniform names at the manuscript.

If some standard or procedure was used, it is necessary to explain - what it is in a few sentences. The reader does not need to be familiar with all standards.

Line 152 - "sample", detailed description, dimensions, etc.

Line 157/159 - SEM / Zwick- model, manufacturer according to the MDPI template.

The description of the tensile tests is not suitable for publication - lack of basic information about the conducted measurements (forces, test conditions etc.).

Table 1. Completely unreadable. It should be widened, the MDPI template allows it.

Picture captions should start with: Image of ..., Picture of ... Map ...

Line 320 - 332 - not needed

Line 334 - an invalid form of citation

Only after correcting the Material and Method section, it will be possible to evaluate the Discussion section reliably. Unfortunately, in the current manuscript state is not possible.

Best regards,

Reviewer

Author Response

Answers to your comments in the attached file.

Reviewer 2 Report

Dear authors:

I read with great interest.  The results could be used by lots of persons.  However, the scientific expressions and discussions seems to be weak, particularly in the light of polymer science. 

Concretely speaking, would you mind revising and modifying the following points?

1: Please insert the scales for all of photos.  Or you can insert something for readers to understand the dimensions and sizes.

2: Would you mind adding the diagrams of the results for gene analyses?  I believe you could show the bar graphs for the components of bacterial flora.  

3: Then you could show PCoA analyses and Jaccard distance in addition.  Then readers could understand the contents more.  At this point, you showed just tables.  I am afraid that readers would have difficulty to follow your great contents.  If you could show those figures (flora and PCoA), your paper would look more scientific.  

4. The explanation for materials is insufficient.  You should explain more in detail what kind of composite materials you used.  And you should also explain why you used basalt plastic reinforcement.  I strongly feel that such an explanation and description would lead readers to their own ideas etc.  Biofouling would occur by the combination between materials and organisms.  Why did they interact each other?  The explanation depends on the mutual forces, so I believe.  To discuss about the topic, readers would need the information on materials sides.  

Author Response

(The authors gave the same response as above.)

Round 2

Reviewer 1 Report

The comments and requirements were addressed, and the manuscript was revised well. 

Reviewer 2 Report

Dear authors. 

I confirmed that you revised your original manuscript faithfully and sincerly according to reviewers' comments and requirements.  I would say that your future publication would attract lots of readers.